# Analysis and prediction of the major fatty acids in vegetable oils using dielectric spectroscopy at 5–30 MHz

Masyitah Amat Sairin[1], Samsuzana Abd Aziz[1,2], Chan Yoke Mun[3], Alfadhl Yahya Khaled[4], Fakhrul Zaman Rokhani[1,5]*

1 Faculty of Engineering, Universiti Putra Malaysia (UPM), Selangor, Malaysia, 2 Smart Farming Technology Research Center, Universiti Putra Malaysia (UPM), Selangor, Malaysia, 3 Faculty of Medicine and Health Sciences, Universiti Putra Malaysia (UPM), Selangor, Malaysia, 4 Department of Horticulture, College of Agricultural & Life Sciences, University of Wisconsin, Madison, WI, United States of America, 5 MyAgeing Research Institute, Universiti Putra Malaysia (UPM), Selangor, Malaysia

* fzr@upm.edu.my

## Abstract

A dielectric spectroscopy method was applied to determine major fatty acids composition in vegetable oils. Dielectric constants of vegetable oils were measured in the frequency range of 5–30 MHz. After data pre-treatment, prediction models were constructed using partial least squares (PLS) regression between dielectric spectral values and the fatty acids compositions measured by gas chromatography. Generally, the root means square error of validation (RMSECV) was less than 11.23% in the prediction of individual fatty acids. The determination coefficient ($R^2$) between predicted and measured oleic, linoleic, mono-unsaturated, and poly-unsaturated fatty acids were 0.84, 0.77, 0.84, and 0.84, respectively. These results indicated that dielectric spectroscopy coupled with PLS regression could be a promising method for predicting major fatty acid composition in vegetable oils and has the potential to be used for in-situ monitoring systems of daily consumption of dietary fatty acids.

## Introduction

The consumption of oils and fats, among other food components play an important role in maintaining a balanced and healthy diet for humans. Fat is essential in the human diet attributed to its energy density used during the day to survive for many weeks without food [1]. Other than that, oil and fat play their essentiality for physiological function, growth, and development, including as a structural component in cell membranes, and act as a carrier for fat-soluble components such as vitamins A, D, E, and K. Besides their importance in providing nutrition for the human body, fat and its lipid components have an important role in food processing: functioning in heat transfer, enhancing food flavor and texture, and characterizing many food products [2].

Globally, the correspondence burden on health care system accompanying the consistent increase of prevalence of cardiovascular signifies the need for a healthier diet. Despite

**Data Availability Statement:** All relevant data are within the manuscript and its Supporting Information files.

**Funding:** Funding was provided by the Malaysian Ministry of Higher Education (www.mohe.gov.my) through the Fundamental Research Grant Scheme under the Grant No. [01-01-20-2247FR], awarded to FZR. The funders had no role in study design, data collection and analysis, decision to publish, or preparation of the manuscript.

**Competing interests:** The authors have declared that no competing interests exist.

inconsistency exists, diets comprised of higher poly-unsaturated fatty acids (PUFA) and mono-unsaturated fatty acids (MUFA) intake are often associated with lower cardiovascular disease risk as compared to diets rich in saturated fatty acids (SFA). This has contributed to the world consumption of major fats and oils being dominated by vegetable oil, i.e., soybean oil, canola oil, and sunflower oil [3], which have a relatively higher composition of unsaturated fatty acids than SFA. This highlights the importance of investigating vegetable oils' fatty acids composition for consumers, food manufacturers, and food authorities. Major fatty acids such as C18:1, C18:2, and C18:3 are abundant in vegetable oils [4, 5]. FAO/WHO (1994) recommends an n-6:n-3 FA ratio of 5–10 [6], while the n-6:n-3 fatty acid ratio of a chemically analyzed typical Malaysian diet was about 24.2±9.6 (~8.45±0.64/0.45±0.56g) (Karupaiah et al., 2016), suggesting a significant deviation from the optimal recommendation [7]. Cell membranes that contain high PUFA are the most stable and permeable to water, thus ensuring efficient intracellular metabolism [8]. It has been reported that the intake of essential fatty acids (EFA) such as C18:2 and C18:3 may have a role in preventing attention deficit hyperactivity disorder (ADHD) and boosting learning skills [9].

Several lab-based approaches have been used for fatty acids composition analysis in past studies. Some conventional approaches include the widely used chromatographic methods like gas chromatography flame ionization detector (GC-FID) [10], gas chromatography hyphenated with time-of-flight (GC-ToF) [11], gas chromatography-mass spectrometry (GC-MS) [12, 13] and high-performance liquid chromatography (HPLC) [14, 15]. While these conventional methods accurately reflect the fatty acids composition of oils, they are time-consuming, involve laborious sample preparation, and are limited to lab-based investigation which requires skilled operators. Hence, developing a simpler, rapid, and portable method for in-situ inspection is warranted.

More recently, a number of spectroscopic methods were introduced for fatty acids characterization in edible oils, such as nuclear magnetic resonance (NMR) [16, 17], Fourier transform infrared (FTIR) [18, 19], Raman spectroscopy [5, 20], near-infrared (NIR) spectrometry [21–23], and impedance spectroscopy [24] (Table 1). These methods utilized statistical approaches, namely least squares—support vector machine (LS-SVM), partial least squares (PLS), and modified partial least squares (MPLS) and could predict food's fatty acid composition with up to 99% accuracy. These methods correlate fatty acid composition with different oil behavior across the electromagnetic spectrum for fatty acids characterization. Generally, different fatty acid compositions lead to different parameters' behavior across the electromagnetic spectrum [25]. Consequently, this spectroscopic characterization capability is instrumental for developing a fatty acids composition prediction model in sensor development [26].

Recently, the potential of using dielectric spectroscopy in the characterization of fatty acids composition in vegetable oil has drawn the attention of scientists [25, 34, 35]. In comparison to the NMR technique, which observes the interaction of magnetic fields with atomic nuclei, dielectric spectroscopy investigates the interaction of electromagnetic radiation with atoms or molecules [36, 37]. Unlike infrared spectrum used in Raman (0.3 THz—120 THz) [38], FTIR (12–120 THz) [39], and NIR spectroscopy (118–384 THz) [21], dielectric spectroscopy on vegetable oil usually covers at a relatively much lower frequency which constitutes at microwaves and radio waves (5–30 MHz); thus, the cost is fairly inexpensive for industrial implementation [40]. In this research, dielectric spectroscopy was used to investigate the variation of dielectric spectra of vegetable oil and to predict the fatty acids composition in the oil at 5–30 MHz. Lower frequency range (i.e., 500Hz– 1MHz) has been studied [26]. Our work complements the current body of knowledge by covering higher frequency range.

The study aims to find the correlation between the major fatty acids' composition such as C18:1, C18:2, C18:3 and MUFA, PUFA and SFA and the dielectric spectral measurements; and

**Table 1. Methods for the prediction of fatty acids.**

| Fatty acid investigated | Sample | Method | References |
|---|---|---|---|
| **Raman Spectroscopy** | | | |
| 16:0, 18:0, 18:1, 18:2, 18:3 | EVOO, rapeseed, peanut, camellia, soybean, sunflower, corn oils | LS-SVM | [20] |
| 16:0, 18:0, 18:1n-9, 18:2n-6, 18:3n-3, 20:1n-9, 22:1n-9, 24:1n-9 | Olive, palm, sunflower, corn, canola, soybean, and mustard oils | PCA and PLS | [5] |
| **Fourier Transform Infrared Spectroscopy (FTIR)** | | | |
| EPA (20:5), DHA (22:6), SFA, MUFA, PUFA | Marine oil omega-3 supplements | PLS | [27] |
| 18:1, 18:2, SFA, MUFA, PUFA | VOO | PLS | [28] |
| 14:0, 14:1, 16:0, 16:1, 18:0, 18:1, 18:2, 18:3, 20:0, 20:1, 20:2, 20:3, 20:4, 20:5, 22:0, 22:1, 22:6, 24:0, 24:1, SFA, MUFA, PUFA, EPA + DHA | Fish fillets | PCA and PLS | [29] |
| **Near-Infrared Reflectance (NIR)** | | | |
| 16:0, 18:0, 18:1, 18:2, 18:3, 20:0, 22:0, (EPA) 20:5, SFA, unsaturated fatty acids | Sunflower seeds oil | MPLS and PLS | [26] |
| 14:0, 16:0, 18:0, 18:1, 18:2, 18:3, SFA, MUFA, PUFA | Ham | PCA and PLS | [30] |
| 4:0, 6:0, 8:0, 10:0, 12:0, 14:0, 16:0, 18:0, 18:1, 18:2, 18:3, trans fatty acid | Fresh and freeze-dried cheeses | PLS | [31] |
| **Near-Infrared Transmittance (NIT)** | | | |
| 10:0, 12:0, 13:0, 14:1, 15:0, 16:0, 16:1, 17:0, 18:0, 18:1, 18:2, 18:3, 19:0, 20:0, 20:2, 20:3, 20:4, 22:0, 22:4, 22:5, 22:6 | Beef | MPLS | [32] |
| 16:0, 18:0, 18:1, 18:2, unsaturated fatty acids | Camellia oil | PLS | [33] |

EVOO: extra virgin olive oil, LS-SVM: least squares—support vector machine, PCA: principal component analysis, PLS: partial least squares, EPA: eicosapentaenoic acid, DHA: docosahexaenoic acid, SFA: saturated fatty acids, MUFA: mono-saturated fatty acids, PUFA: poly-unsaturated fatty acids, VOO: virgin olive oil, MPLS: modified partial least squares.

to utilize the spectral measurements for quantitative prediction of the fatty acids as an alternative to chromatographic analyses. As major fatty acids mostly found in human diets are the 16 and 18 carbon chains, this study focuses on C18 fatty acids [41].

## Materials and methods

### Oils sample preparation

Prior to the administration of the study, a market survey was conducted to identify the availability of various brands of cooking oils available locally. To the best of our knowledge, there were five types/brands of olive, canola, sunflower oils, respectively, and two types/brands of soybean oils available in the market at the time of data collection. A total of 17 samples were selected based on their availability in the market as well as their popularity [42–44]. The samples were procured from local markets in Speedmart (N 3.01008°, E 101.71380°), Aeon Supermarket (N 2.995400°, E 101.67512°), and Jaya Grocer (N 2.92736°, E 101.65041°), Selangor, Malaysia. All samples were sold in bottles. Oil samples were selected from typical edible oils according to their fatty acids type, i.e., oleic acid (olive and canola oil); and linoleic acid (soybean and sunflower oil) [20, 25]. All oil samples were kept in amber bottles inside a dark storage box at room temperature (23–25°C) before analysis.

### Fatty acid methyl ester (FAME) preparation

Fatty acid methyl esters (FAME) were derived from the oil samples according to Japan Oil Chemist's Society (JOCS) Standard Method 2.4.1.3–2013 [45]. A total of 50 mg of oil was dissolved in 0.8 ml of hexane, and 0.2 ml 1M sodium methoxide was added. The mixture was

vortexed for 1 minute for the hydrolysis process and derivatization reactions to take place. The clear supernatant upper layer of the solution was transferred into a 2 ml vial before GC-MS analysis. Standard 37 FAME compounds (C4 to C24) were diluted with hexane to prepare the standards for GC-MS measurement. Standard FAME of 37 compounds (C4 to C24) was purchased from (Supelco, Sigma-Aldrich, Bellefonte, PA). Chromatography grade of n-hexane and sodium methoxide were purchased from (Merck Chemicals, Darmstadt, Germany). All chemicals used in the experiment were analytical grade.

### Gas chromatography mass spectrometry (GC-MS) measurement

The fatty acid compositions were determined using a 7890A gas chromatography equipped with 5975 mass spectrometry and a polar capillary column HP88 with 0.25 mm internal diameter, 100 m length, and 0.25 μm film thickness available from Agilent Technologies, USA. Using helium as a carrier gas at a flow-rate of 0.8 ml/minute, samples were injected with a split ratio of 30:1. The initial temperature was set at 150°C for 5 minutes and programmed to increase to 240°C at 4°C/minute. The final temperature was maintained at 240°C for 15 minutes. The run time for one injection was 42.50 minutes. The analysis was conducted at three injection replications with randomized order of oil samples. The FAME peaks were identified by comparing their retention time with certified FAME reference standards (Supelco, Sigma-Aldrich, USA). The percentage of fatty acid was calculated based on the peak area ratios of a fatty acid species to the total peak area of all the fatty acids in the oil sample. The results were expressed in percentage (%).

### Dielectric spectroscopy measurement

Dielectric constant (ε') of 3.4 ml oil samples was measured with a randomized order of oil samples using a liquid dielectric test fixture (Agilent 16452A, Agilent Technologies, Hyogo, Japan) that is affixed to a precision impedance analyzer (Agilent 4294A, Agilent Technologies, Hyogo, Japan) using 16048G test leads (Agilent Technologies, Hyogo, Japan). For control and data logging, a personal computer is attached to the aforementioned measurement system. The ε' measurement of the oil sample provides a measure of its effect on the ratio of the capacitance of a capacitor containing the oil sample to that air capacitance. Prior to the measurement, calibration of the liquid test fixture was performed using air, distilled water, methanol, and ethanol at 100kHz for error correction following the standard procedure according to the instrument operation manual [46]. The ε' values from this study and Dortmund data bank [47–50] were plotted as Fig 1. The ε' values change between 1.00035–1.00059, 71.6–73.4, 28.4–30.4, and 21.3–22.4 for air, distilled water, methanol, and ethanol, respectively. The high ε' values of distilled water are because the water molecule has a dipole moment, so it can be polarized. Under a given electric field, water tends to align with the field, thus, it's polarized strongly.

The ε' was measured in the frequency range of 5 MHz to 30 MHz at 166 discrete frequencies with 150 kHz intervals, compatible with the dielectric test fixture. To ensure all of the samples were liquid and homogenous, the dielectric measurements were conducted in a temperature-controlled chamber at 45 ± 0.1°C (Espec SU221, Michigan, USA) [51, 52]. The measurement was saved after three replications. For data analysis, the average of three replications is calculated. The liquid test fixture was cleaned after each measurement.

### Data pre-processing

The dielectric spectral data were normalized using the mean normalization method in the Unscrambler X software version 10.4 (CAMO Software AS, Trondheim, Norway). Mean normalization is where each row (sample) of a data matrix is divided by its average. Normalization

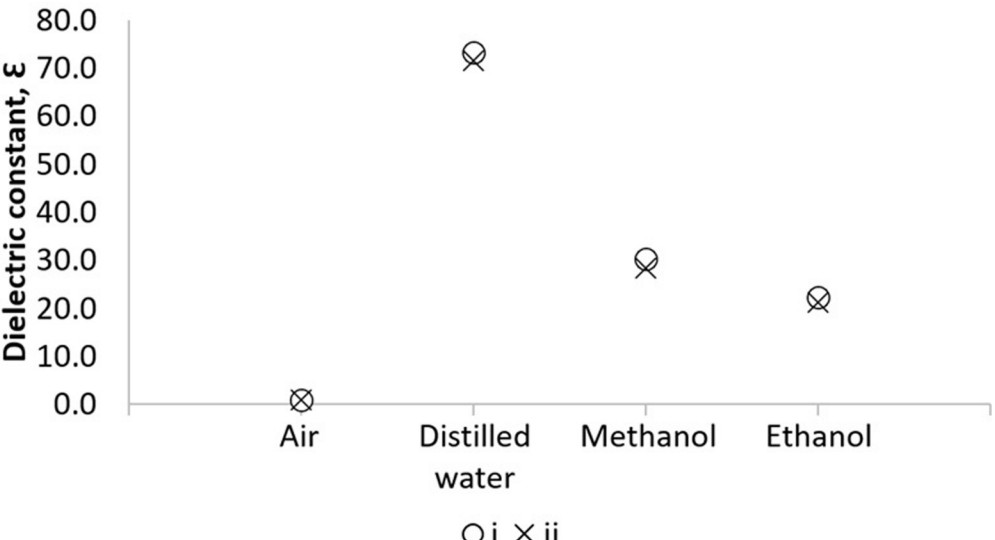

**Fig 1.** Comparison of calibration results using air, distilled water, methanol, and ethanol between the data from (i) this study and (ii) Dortmund Data Bank.

is an essential data pre-processing method [53] that effectively scales the dielectric measurements across the frequencies [54].

## Statistical analysis and model development

Data were analyzed using Minitab software, Release 16 (Minitab Inc., State College, PA, USA) statistical package, with $p < 0.05$ was set as a significant level. Two-way analysis of variance (ANOVA) was performed to determine the effect of independent variables (e.g., frequency and oil group) on mean dielectric constant, while the Tukey test with one-way ANOVA was utilized to compare the means of dielectric constant and fatty acids composition of different vegetable oils.

Principal component analysis (PCA) was utilized to explore the influence of each fatty acid on the variation of the GC-MS and spectral data. Then, principal component regression (PCR) and partial least squares (PLS) regression in Unscrambler X software version 10.4 (CAMO Software AS, Trondheim, Norway) was used for model development to predict the fatty acids compositions in the oil. Full cross-validation of the model was conducted where one sample was left out at a time from the calibration data set. The value of the left-out sample is predicted, and the prediction residual is computed. The process was repeated with another sample until every sample had been left out once. The root means square error of cross-validation (RMSECV), mean absolute percentage error (MAPE), and determination coefficients, $R^2$ was used to measure model performance. The RMSECV and MAPE were calculated using Eqs 1 and 2, respectively:

$$RMSECV = \sqrt{\frac{1}{N}\sum_{i=1}^{N}(X - Y)^2} \qquad (1)$$

$$MAPE = \frac{100\%}{N}\sum_{i=1}^{N}\left|\frac{X - Y}{X}\right| \qquad (2)$$

where N is the number of samples, Y is the predicted measurement using the calibration model, and X is the left-out sample value.

## Results and discussion

### GC-MS results and fatty acids evaluation

Table 2 depicts the fatty acids composition of vegetable oils as acquired using GC-MS analysis. The major fatty acid of canola oil and olive oil comprised 57.53% and 83.26% C18:1 (oleic acid), respectively, while sunflower oil and soybean oil comprised 60.34% and 62.72% of C18:2 (linoleic acid), respectively. These FAME profiles confirmed the fatty acids classifications observed in another study [20].

Canola oil contains approximately 15% C18:3 (linolenic acid), which was significantly higher than olive, sunflower, or soybean oils. On the other hand, the C18:3 content in olive oil (2.79%) and sunflower oil (1.41%) was significantly lower than soybean oil ($p < 0.05$). Olive oil contains approximately 84% of MUFA, which was significantly higher than the other vegetable oils, while the PUFA content in soybean oil (73.69%) and sunflower oil (64.24%) were significantly higher than the other vegetable oils (Table 2).

Fig 2A score plot shows that each oil sample was located on a different area according to their vegetable oil group and showed four well-defined and well-separated groups.

The loading plot in Fig 2B is used to determine which fatty acids influence the separation of the oil samples. The absolute value of the loading in a principal component describes the importance of the contribution of the particular component. The further a variable is away from the origin, the greater the contribution of that specific variable to the principal component [55]. As shown in Fig 2B, the main compounds that caused the separation of the samples were C18:1, C18:2, C18:3, MUFA, and PUFA. The high positive correlation between C18:2 and PUFA along PC-1 indicated that soybean and sunflower oil profiles have a higher proportion of C18:2 and PUFA. On the other hand, the high negative of MUFA and C18:1 along PC-1 indicated a higher proportion of the compounds in olive oil. Last but not least, the sole highest negative of C18:3 along PC-2 indicated a higher proportion in canola oil.

For the vegetable oil samples in this study, C18:2-PUFA and C18:1-MUFA are negatively correlated, indicated by the angles between their vectors close to 180° as explained in the previous study [56]. This further indicates that when the proportion of C18:2-PUFA increases, C18:1-MUFA decreases. From GC-MS analysis (Table 2), C18:1, C18:2, and C18:3 was shown to be the highest composition (%), and it was further proven as a highly significant statistically in PCA analysis.

**Table 2. Composition of major fatty acids, SFA, MUFA, and PUFA in vegetable oils.**

| Fatty acid | Weight (%) | | | |
|---|---|---|---|---|
| | Olive | Canola | Soybean | Sunflower |
| 18:1 | 83.26 ± 6.40[a] | 57.53 ± 4.92[b] | 25.90 ± 0.52[c] | 35.49 ± 5.89[c] |
| 18:2 | 12.56 ± 6.30[c] | 24.81 ± 6.12[b] | 62.72 ± 0.91[a] | 60.34 ± 6.12[a] |
| 18:3 | 2.79 ± 0.27[c] | 15.12 ± 1.81[a] | 9.49 ± 0.36[b] | 1.41 ± 0.26[c] |
| SFA | 0.27 ± 0.21[b] | 0.69 ± 0.05[a] | 0.19 ± 0.01[b] | 0.14 ± 0.02[b] |
| MUFA | 83.69 ± 6.30[a] | 57.69 ± 4.92[b] | 26.12 ± 0.47[c] | 35.62 ± 0.47[c] |
| PUFA | 16.03 ± 6.39[c] | 41.61 ± 4.95[b] | 73.69 ± 0.47[a] | 64.24 ± 0.47[a] |

Note: 1. All determinations were carried out in triplicate, and mean ± SD were reported.

2. Mean weight (%) values not followed by a common lower-case letter (a, b, c) differ significantly ($p<0.05$).

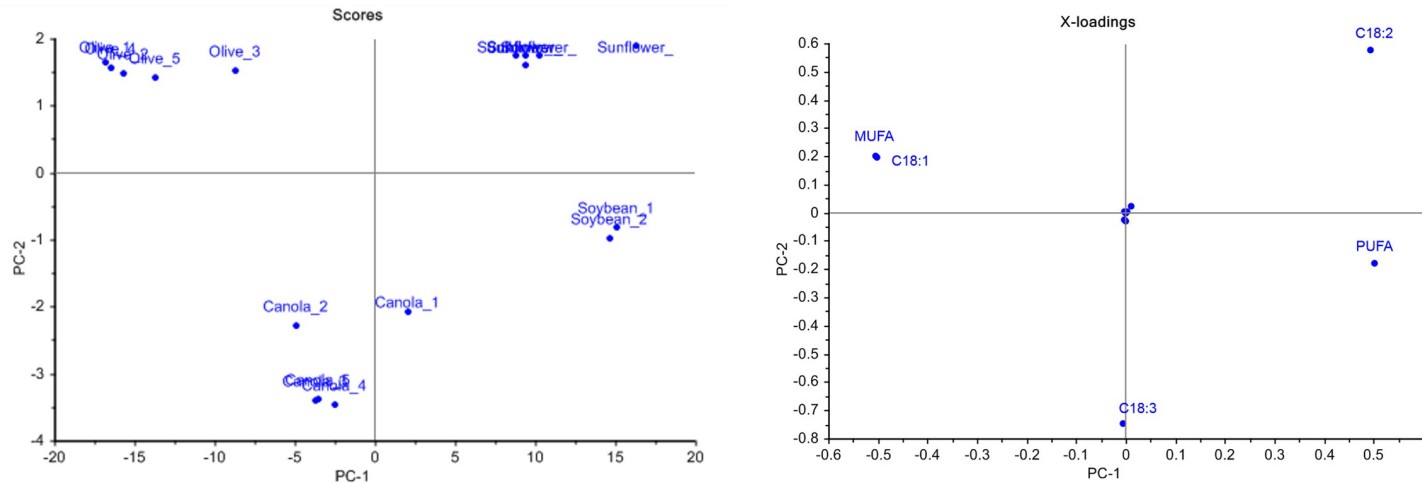

**Fig 2.** PCA (a) score plot of the analyzed oil samples and corresponding (b) loading plot of fatty acids, obtained by GC-MS.

## Dielectric constant distribution of vegetable oils

Fig 3 exhibits the mean dielectric constant (ε') spectra for individual oil groups (i.e., canola oil, olive oil, sunflower oil, and soybean oil). In general, the dielectric spectra had an inverse correlation with the frequency of the electric field, where the dielectric constant decreases with the increase of frequency. This behavior was observed on all vegetable oil samples. According to Lizhi *et a.l*, the dielectric constant diminishes in higher frequency [25]. Datta *et al*. mentioned that at high frequency, the dipoles of dielectric materials are incapable of following the shorter field reversal and are not reacting to the electric field [57]. From Table 3, the two-way ANOVA showed that the frequency and oil group were the significant factors affecting dielectric properties.

Across the spectra, soybean oil had the highest mean ε', while olive oil exhibited the lowest ε' values. The mean ε' of soybean oil and sunflower oil overlapped at low frequency, and the mean ε' of canola oil and sunflower oil overlapped at mid-range frequency. To further confirm

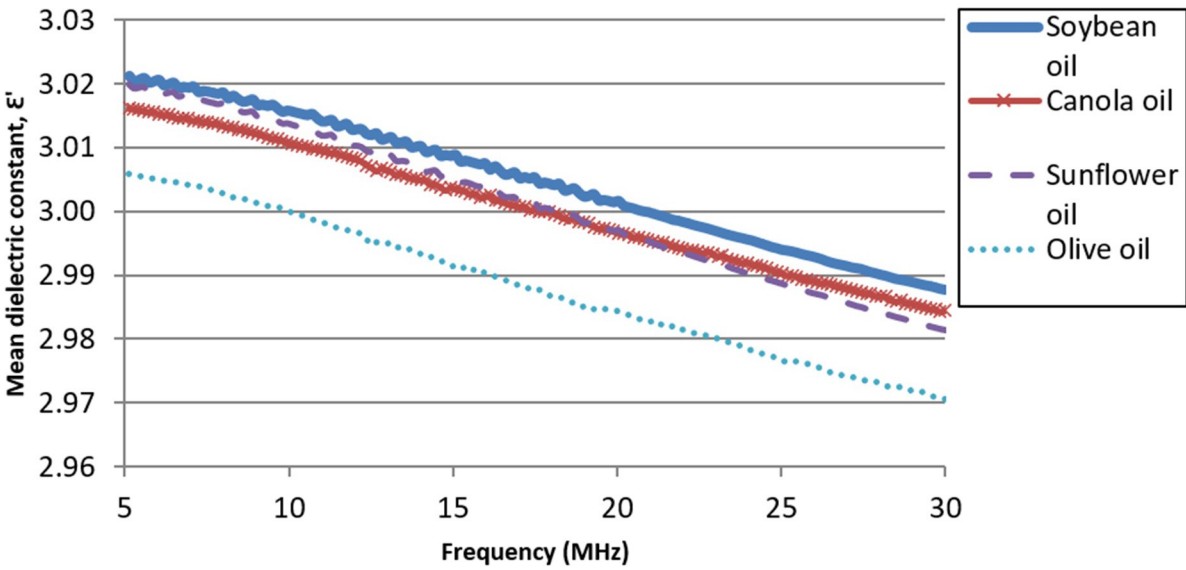

**Fig 3. Mean dielectric constant spectra of different vegetable oil groups.**

**Table 3. Two-way ANOVA of the mean dielectric constant of vegetable oils.**

| Dependent Variable: Mean dielectric constant | | | | | |
|---|---|---|---|---|---|
| Source | Degree of freedom | Sum of squares | Mean square | F-value | p-value |
| Frequency | 165 | 0.07 | 0.0004 | 513.60 | < .005 |
| Oil group | 3 | 0.02 | 0.0085 | 9864.20 | < .005 |

the difference of ε' between oil groups statistically, the Tukey test was performed on the means of the ε' of each oil across all frequencies, as shown in Table 4. In general, there were no significant differences (p<0.05) between the mean measured ε' of sunflower and canola oil, but the mean measured ε' of soybean and olive oil were significant different from those obtained from sunflower and canola oils.

The ε' of vegetable oils were generally affected by unsaturated fatty acids of C18, which were C18:1, C18:2, and C18:3 [25], whereby the higher the number of double bonds present in the carbon chain, the more unsaturated the oil is. Oils of oleic (C18:1) type, i.e., contain a higher amount of oleic acid, was observed to have a lower ε' value. On the other hand, oils that belong to linoleic (C18:2) type, i.e., contain a higher amount of linoleic acid and have a higher degree of unsaturation, exhibited a higher ε' value. In addition, a higher degree of unsaturation of the fatty acid means having more double bonds and fewer hydrogen atoms, leading to having a lower molecular weight [58], which makes the molecules easily aligned to the electric field, thus resulting in slightly higher ε'.

## Correlation and prediction of fatty acids composition with dielectric spectra

Fig 4 compares the distributions of ε' across four different groups of vegetable oils and their correlations with individual fatty acids. The box plot of soybean oil shows long whiskers at the bottom, indicating the underlying distribution is skewed toward the lower dielectric constant. C18:2 and PUFA show a positive correlation with ε'; C18:1 and MUFA show a negative correlation with ε'; while no apparent correlation was observed in C18:3 and SFA.

Fig 5 displays the PCA correlation loading plot for vegetable oils, showing how strong each fatty acid, SFA, MUFA, PUFA, and ε' variable influences a principal component. All the points on the plot form the loading vectors of each fatty acid corresponding to the origin. Loadings close to -1 or 1 indicate that the variable strongly influences the component. The loadings between the two circles of the plot represent 50–100% significance of the variable's influence. Based on the correlation loading plot, it was found that the ε' had a significant influence on PC-1 and PC-2, compared to the fatty acids composition. Among the fatty acids, C18:1, C18:2, MUFA, and PUFA showed larger loading vectors on PC-2, indicating a stronger influence on PC-2.

On the other hand, C18:3 and SFA had a weaker influence on PC-2, which loadings closer to zero. It is also observed that ε' is negatively correlated to C18:1 and MUFA composition, while positively correlated to C18:2 and PUFA. This opposite relation towards dielectric

**Table 4. Tukey test on the mean dielectric constant of vegetable oil.**

| Samples | Mean dielectric constant, ε' |
|---|---|
| Soybean oil | $3.0048 \pm 0.0102^a$ |
| Sunflower oil | $3.0009 \pm 0.0096^b$ |
| Canola oil | $3.0004 \pm 0.0117^b$ |
| Olive oil | $2.9882 \pm 0.0108^c$ |

*Mean dielectric constant values not followed by a common lower-case letter (a, b, c) differ significantly (p<0.05).

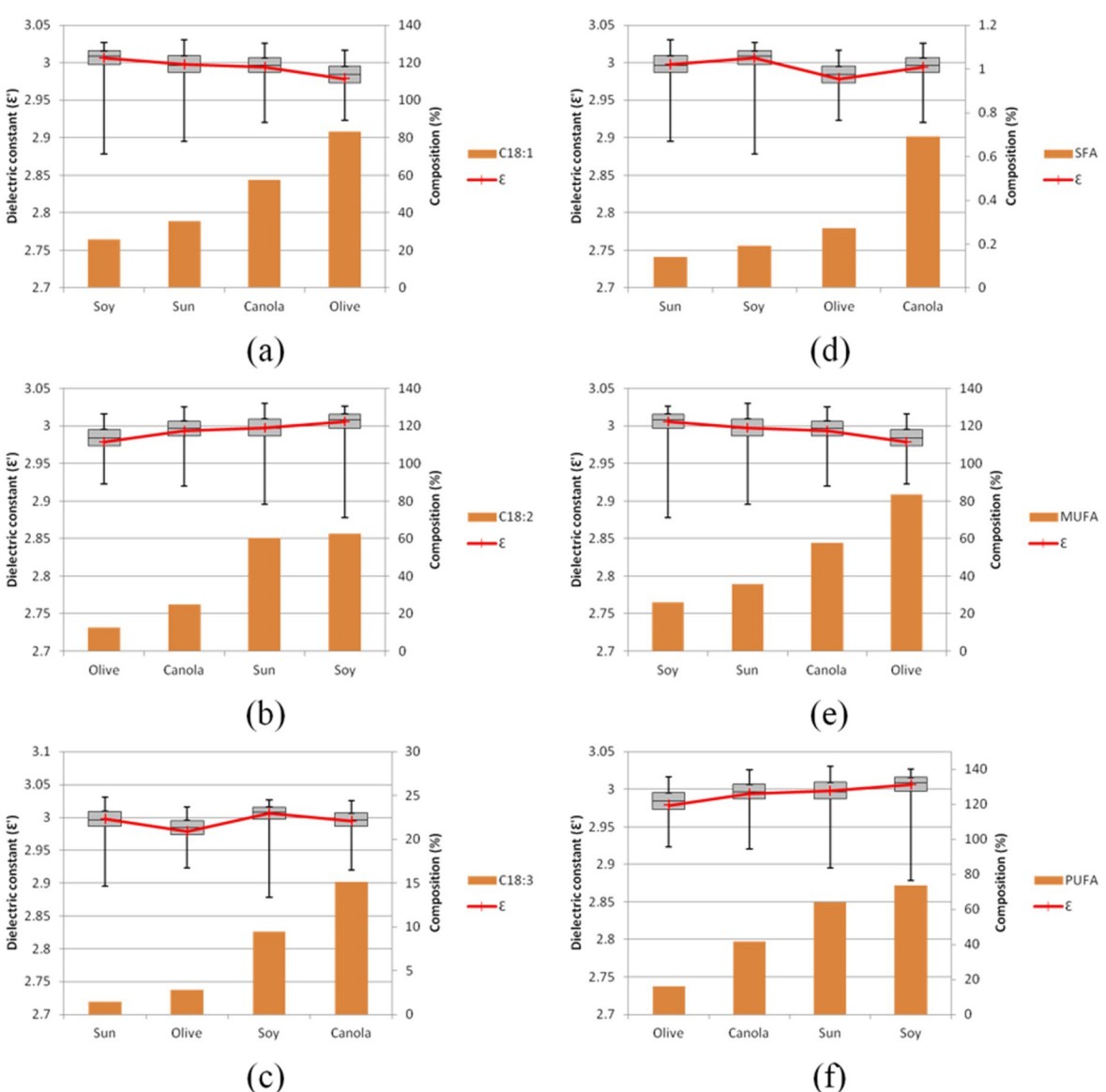

**Fig 4.** Correlation of (a) C18:1, (b) C18:2, (c) C18:3, (d) SFA, (e) MUFA and (f) PUFA composition with ε'.

constant was indicated as the two groups (C18:1-MUFA and C18:2-PUFA) were at two extreme ends of PC-2. As shown in Fig 5, the PCA loading plot, C18:3, and SFA correlate less to ε'. The findings from the correlation loadings agree with the results from Fig 4.

To predict the fatty acids profile (C18:1, C18:2, C18:3, SFA, MUFA, and PUFA) in vegetable oils, calibration models were developed using PCR and PLS regression algorithms with the pre-processed dielectric spectral data. The dependent variables, Y, were fatty acid components (C18:1, C18:2, C18:3, SFA, MUFA, and PUFA) of the vegetable oils, while the independent variables, X, were ε' of oils measured over 166 frequencies. The optimal number of latent variables and principal components for the PLS and PCR regression model for all fatty acid models was determined to be five, where the cumulative explained variance is high, and the RMSECV is at the minimum as suggested in a previous study [59]. PCR and PLS regression showed substantial cross-validation performance as measured by RMSECV for all fatty acids, as shown in Table 5.

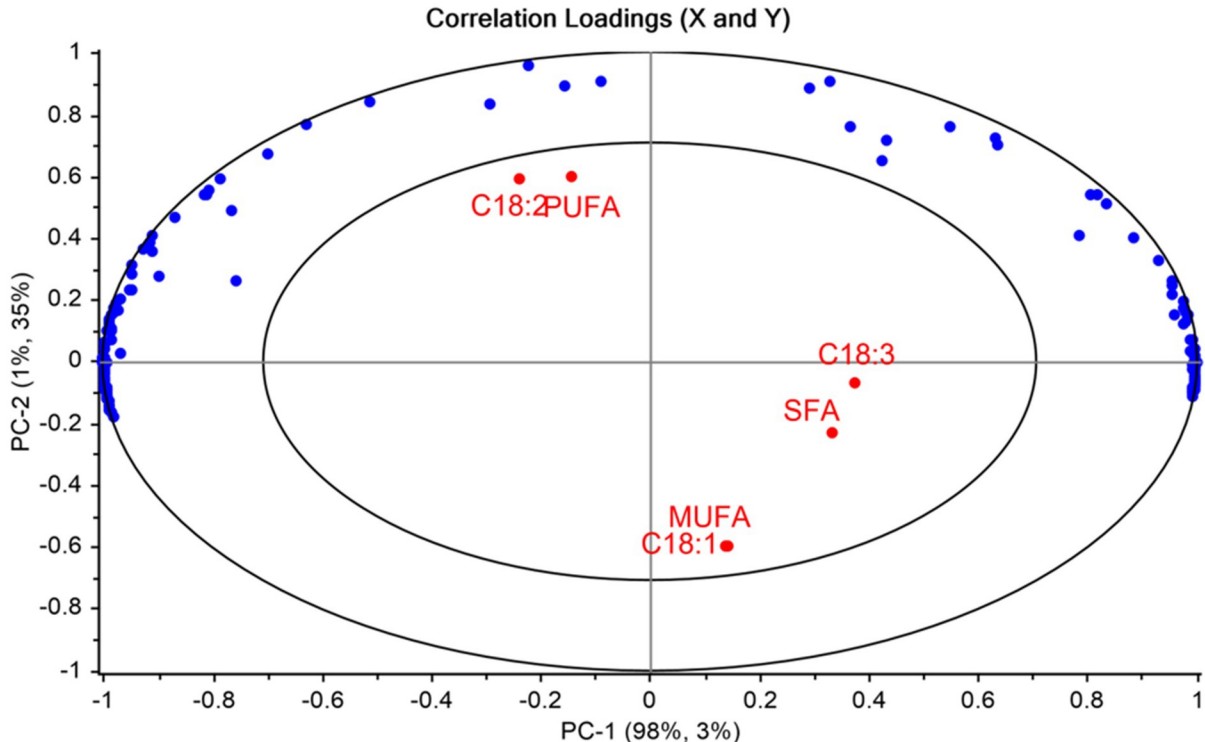

**Fig 5. PCA correlation loading plot for major fatty acids, SFA, MUFA, and PUFA of vegetable oils and dielectric constant.**

The PCR models gave coefficient of determination ($R^2$) values between 0.39 to 0.84 for individual fatty acids. C18:1, MUFA, and PUFA showed the most accurate results relating the predicted composition to the measured composition, with an $R^2$ of 0.84, and MAPE of 18.05% to 18.90%. This is followed by C18:2 with an $R^2$ of 0.77 with slightly higher MAPE of 23.5%. Fig 6 shows the predicted values versus measured values of the PCR regression model for MUFA. The dielectric prediction model for these fatty acids' composition gives the most accurate results because of their higher percentage of composition in the oil.

Poor $R^2$ and very high MAPE were observed for C18:3 and SFA due to low percentage composition in oil. Most vegetable oils contain less than 16% of C18:3, and less than 1% of SFA with reference to the total fatty acid composition, as displayed in Table 2. This echoed earlier studies whereby Akkaya *et al*. and Yuan *et al*. also recorded relatively low correlation for C18:3 and SFA [26, 33]. C18:3 and SFA are a small fraction of lipids, and their content is somewhat similar in a variety of vegetable oil.

**Table 5. Prediction evaluation of the PCR and PLS analysis.**

| Fatty acids | PLS Model | | | | PCR Analysis | | | |
|---|---|---|---|---|---|---|---|---|
| | No. Latent Variables | RMSECV (%) | MAPE (%) | $R^2$ | No. Principal Component | RMSECV (%) | MAPE (%) | $R^2$ |
| C18:1 | 5 | 9.19 | 18.08 | 0.84 | 5 | 9.21 | 18.22 | 0.84 |
| C18:2 | 5 | 11.23 | 23.58 | 0.77 | 5 | 11.19 | 23.50 | 0.77 |
| C18:3 | 5 | 5.28 | 154.88 | 0.40 | 5 | 5.29 | 157.32 | 0.39 |
| SFA | 5 | 0.18 | 55.14 | 0.55 | 5 | 0.18 | 56.36 | 0.55 |
| MUFA | 5 | 9.17 | 17.92 | 0.84 | 5 | 9.19 | 18.05 | 0.84 |
| PUFA | 5 | 9.24 | 18.85 | 0.84 | 5 | 9.26 | 18.90 | 0.84 |

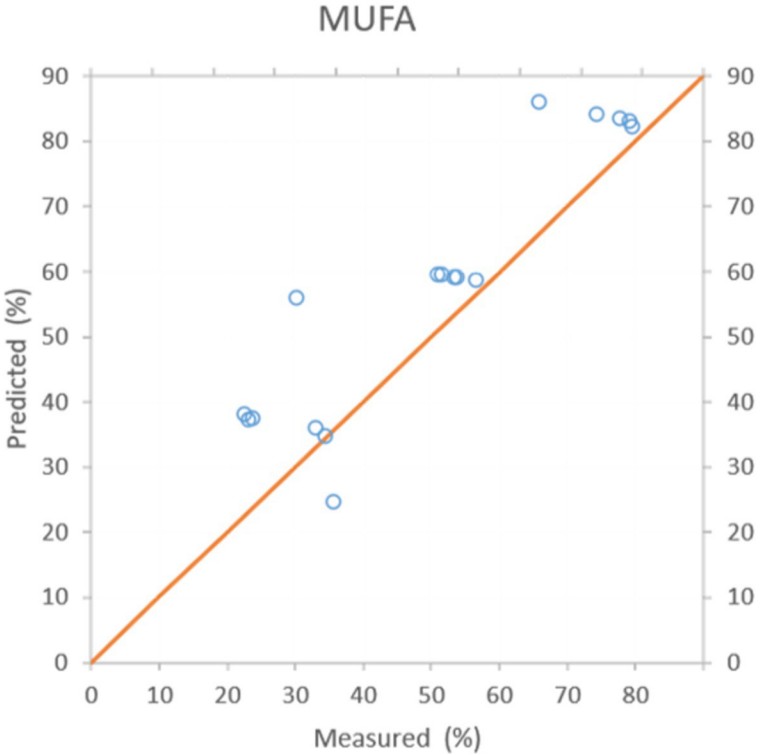

**Fig 6. Predicted versus measured for MUFA using PCR regression model ($R^2$ = 0.84).**

The PLS models developed by correlating dielectric constant spectra to fatty acid composition gave MAPE values ranging from 17.92% to 23.58%. Among the best $R^2$ achieved in PCR, the best RMSECV result obtained was for MUFA, with a value of 9.17%. Although RMSECV for C18:3 and SFA are much lower at 5.28 and 0.18, respectively, these numbers are relatively high compared to their range of composition in the oil. These are further supported by the MAPE values of 154.88% and 55.14% for C18:3 and SFA, respectively. Further inspection reveals that the absolute percentage errors are relatively high in sunflower oil samples due to the smallest composition of C18:3 and SFA compared to other vegetable oils. This is due to the high similarity of sunflower and canola oil dielectric spectra as shown in Fig 3 and the very low composition of C18:3 and SFA in sunflower oil compared to canola oil.

## Conclusion

This study developed a quantitative prediction model of the vegetable oils' major fatty acids, SFA, MUFA, and PUFA composition by utilizing the dielectric spectral data through the PCR and PLS regression analysis. The relationship between the dielectric spectra at 5–30 MHz and fatty acids composition was analyzed. The ε' was negatively correlated to C18:1 and MUFA composition, while positively correlated to C18:2 and PUFA. The predictions for C18:1, C18:2, MUFA, and PUFA showed high correlation coefficients (0.77–0.84), small RMSECV (9.17% - 11.23%) and moderate MAPE (17.92%– 23.58%), while the correlation and prediction of C18:3 and SFA gave lower coefficients. The proposed method offers a rapid and simple technique to assess vegetable oils' nutrition value, facilitating a future study to develop an in-situ oil quality monitoring system to monitor the daily consumption of fatty acids. The proposed

method is a highly practical alternative regarding sample preparation and time without compensating for the accuracy.

## Supporting information

**S1 File.**
(DOCX)

## Acknowledgments

The authors are very grateful to the researchers and staff from the Halal Products Research Institute (HPRI) and the Environmental and Biosystem Lab, Universiti Putra Malaysia (UPM), for assistance in GC analysis and dielectric spectral measurements.

## Author Contributions

**Conceptualization:** Fakhrul Zaman Rokhani.

**Formal analysis:** Masyitah Amat Sairin.

**Funding acquisition:** Fakhrul Zaman Rokhani.

**Investigation:** Masyitah Amat Sairin.

**Methodology:** Masyitah Amat Sairin, Samsuzana Abd Aziz.

**Project administration:** Fakhrul Zaman Rokhani.

**Supervision:** Samsuzana Abd Aziz, Chan Yoke Mun, Alfadhl Yahya Khaled, Fakhrul Zaman Rokhani.

**Validation:** Chan Yoke Mun, Alfadhl Yahya Khaled.

**Visualization:** Masyitah Amat Sairin.

**Writing – original draft:** Masyitah Amat Sairin.

**Writing – review & editing:** Samsuzana Abd Aziz, Chan Yoke Mun, Alfadhl Yahya Khaled, Fakhrul Zaman Rokhani.

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
