## [Decision Letter · Decision Letter 0]

22 Nov 2021

PONE-D-21-31530Analysis and Prediction of the Major Fatty Acids in Vegetable Oils using Dielectric Spectroscopy at 5 – 30 MHzPLOS ONE

Dear Dr. Fakhrul Rokhani,

Thank you for submitting your manuscript to PLOS ONE. After careful consideration, we feel that it has merit but does not fully meet PLOS ONE’s publication criteria as it currently stands. Therefore, we invite you to submit a revised version of the manuscript that addresses the points raised during the review process.

We look forward to receiving your revised manuscript.

Kind regards,

Clara Sousa

Academic Editor

PLOS ONE

Journal Requirements:

2. We understand that you purchased products from local markets for this study. In your Methods section, please provide additional regarding the source of this material. Please provide the geographic coordinates and names of the purchase locations (e.g., stores, markets), if available, as well as any further details about the purchased items (e.g., lot number, source origin, description of appearance) to ensure reproducibility of the analyses.

“The present work was supported by the Malaysian Ministry of Higher Education under the Fundamental Research Grant Scheme [01-01-20-2247FR].”

“Funding was provided by the Malaysian Ministry of Higher Education (www.mohe.gov.my) through the Fundamental Research Grant Scheme under the Grant No. [01-01-20-2247FR], awarded to FZR. The funders had no role in study design, data collection and analysis, decision to publish, or preparation of the manuscript.”

Reviewers' comments:

Reviewer's Responses to Questions

**Comments to the Author**

1. Is the manuscript technically sound, and do the data support the conclusions?

Reviewer #1: Partly

Reviewer #2: Yes

2. Has the statistical analysis been performed appropriately and rigorously? 

Reviewer #1: No

Reviewer #2: Yes

3. Have the authors made all data underlying the findings in their manuscript fully available?

Reviewer #1: Yes

Reviewer #2: Yes

4. Is the manuscript presented in an intelligible fashion and written in standard English?

Reviewer #1: Yes

Reviewer #2: Yes

5. Review Comments to the Author

Reviewer #1: Review from Manuscript ID: PONE-D-21-31530

Summary: The paper describes the use of dielectric spectroscopy and multivariate data analysis to determine fatty acids in vegetable oils

General comments:

The paper is in general well written. However, I have some questions regarding the experimental data and data analysis.

• In the data pre-processing can the authors describe what is the mean normalization? Is it the mean center?

• Figure 1- the PCA has 100% of explain variance, so why do a PCA? PCA is a data reduction method. If 100% variance is used, no data reduction occurred.

• My important concern is regarding the PLS analysis. First 17 samples are very few, in particularly if 5 latent variables are going to be used. Furthermore, the authors used full cross validation that is known to overfit the model, additionally, the model was not validated with external samples. This is a crucial point, therefore is my opinion that the authors need to validate the model to the paper to be published.

• In Table 5 where are the r2od the PLS model? Why no figure like Figure 5 was made for the

Reviewer #2: The manuscript by Sairin et al. reports a prediction of fatty acids in vegetable oils by means of dielectric spectroscopy. The paper is well written and organized, and the significance and importance of dielectric spectroscopy study in this field is well clarified. I would like to recommend it to be published in PLOS ONE after the following points being addressed.

1) In the section of “dielectric spectroscopy measurement”, the authors should mention how they perform the calibration and show the calibration results by using a couple of standard liquids.

2) The authors should explain why the frequency range of 5 – 30MHz is selected. Many oils could exhibit a relaxation behavior in this frequency range, if this happens, dielectric constant in this frequency range will show a decrease with increasing frequency, and the dielectric constant measured in this range cannot represent the static dielectric constant of the sample under study.

6. PLOS authors have the option to publish the peer review history of their article (what does this mean?). If published, this will include your full peer review and any attached files.

Reviewer #1: No

Reviewer #2: No

---

## [Author Response · Author response to Decision Letter 0]

21 Feb 2022

Respond to viewer and editor has been included in separate document named "Response to Reviewers"

---

## [Decision Letter · Decision Letter 1]

21 Mar 2022

PONE-D-21-31530R1Analysis and Prediction of the Major Fatty Acids in Vegetable Oils using Dielectric Spectroscopy at 5 – 30 MHzPLOS ONE

Dear Dr. rokhani,

Thank you for submitting your manuscript to PLOS ONE. After careful consideration, we feel that it has merit but does not fully meet PLOS ONE’s publication criteria as it currently stands. Therefore, we invite you to submit a revised version of the manuscript that addresses the points raised during the review process.

We look forward to receiving your revised manuscript.

Kind regards,

Clara Sousa

Academic Editor

PLOS ONE

Journal Requirements:

Reviewers' comments:

Reviewer's Responses to Questions

**Comments to the Author**

1. If the authors have adequately addressed your comments raised in a previous round of review and you feel that this manuscript is now acceptable for publication, you may indicate that here to bypass the “Comments to the Author” section, enter your conflict of interest statement in the “Confidential to Editor” section, and submit your "Accept" recommendation.

Reviewer #1: (No Response)

Reviewer #2: All comments have been addressed

2. Is the manuscript technically sound, and do the data support the conclusions?

Reviewer #1: Partly

Reviewer #2: Yes

3. Has the statistical analysis been performed appropriately and rigorously? 

Reviewer #1: No

Reviewer #2: Yes

4. Have the authors made all data underlying the findings in their manuscript fully available?

Reviewer #1: Yes

Reviewer #2: Yes

5. Is the manuscript presented in an intelligible fashion and written in standard English?

Reviewer #1: Yes

Reviewer #2: Yes

6. Review Comments to the Author

Reviewer #1: Review from Manuscript ID: PONE-D-21-31530R1

Summary: The paper describes the use of dielectric spectroscopy and multivariate data analysis to determine fatty acids in vegetable oils

General comments:

The paper was improved however there are still some minor corrections to be made.

Table 5 should be the one that is in supplementary material. I do not understand why the authors choose to show a “smaller” table in the paper.

In the conclusion section it is written that the errors are between 9.17% and 11.23%, however these errors are the RMSECV and not the MAPE. Authors should discuss the MAPE errors that are between 17.72% and 23.58%.

Reviewer #2: (No Response)

7. PLOS authors have the option to publish the peer review history of their article (what does this mean?). If published, this will include your full peer review and any attached files.

Reviewer #1: No

Reviewer #2: **Yes: **Zhen Chen

---

## [Author Response · Author response to Decision Letter 1]

5 May 2022

We have addressed the issues regarding the reference list. We have reviewed and updated the reference list as per request. Detailed Table 5 was replaced with additional discussions on MAPE for PCR models and also, we changed MAPE (Mean Absolute Percentage Error) for the discussion for errors instead of RMSECV in the conclusion as well as discussion section.

---

## [Editor Report · Decision Letter 2]

10 May 2022

Analysis and Prediction of the Major Fatty Acids in Vegetable Oils using Dielectric Spectroscopy at 5 – 30 MHz

PONE-D-21-31530R2

Dear Dr. rokhani,

We’re pleased to inform you that your manuscript has been judged scientifically suitable for publication and will be formally accepted for publication once it meets all outstanding technical requirements.

Kind regards,

Clara Sousa

Academic Editor

PLOS ONE
---

## [Editor Report · Acceptance letter]

16 May 2022

PONE-D-21-31530R2 

Analysis and prediction of the major fatty acids in vegetable oils using dielectric spectroscopy at 5 – 30 MHz 

Dear Dr. Rokhani:

I'm pleased to inform you that your manuscript has been deemed suitable for publication in PLOS ONE. Congratulations! Your manuscript is now with our production department. 

Kind regards, 

on behalf of

Dr. Clara Sousa 

Academic Editor

PLOS ONE